# Intercropped Plants Provide a Reservoir of Predatory Mites in Coffee Crop

Júlia J. Ferla [1], Gustavo J. de Araújo [1], Madelaine Venzon [2], Pedro H. M. G. Nascimento [1], Milena O. Kalile [1], Shauanne D. Pancieri [1], André C. Cardoso [1], Elem F. Martins [1], Noeli J. Ferla [3] and Angelo Pallini [1,*]

1   Department of Entomology, Federal University of Viçosa, Viçosa 36570-000, Minas Gerais, Brazil
2   Agriculture and Livestock Research Enterprise of Minas Gerais (EPAMIG),
    Viçosa 36570-000, Minas Gerais, Brazil
3   Laboratory of Acarology, Universidade do Vale do Taquari—Univates,
    Lajeado 95914-014, Rio Grande do Sul, Brazil
*   Correspondence: pallini@ufv.br

**Abstract:** Conservation biological control of pests may be achieved using a variety of integrated strategies based on crop diversification. We investigated whether the insertion of the intercropped plants species (IPS) *Inga edulis*, *Senna macranthera*, and *Varronia curassavica* modified the abundance of mites, their feeding behavior, and the dissimilarity of predator and herbivore mites over a gradient of distance from the IPS on coffee. To accomplish this, we recorded the mite species on coffee plants along transects of 16 m extending from the IPS, including on the IPS. A total of 8946 specimens were sampled. Tenuipalpidae was the most abundant family on coffee, followed by Tydeidae, while Eriophyidae was the most abundant on the IPS, followed by Phytoseiidae. The abundance and richness of mites differed between their feeding behavior and distance. The dissimilarity of predators and herbivores increased along a gradient of distance. Furthermore, the IPS harbored several mite species and the diversity of predator and herbivore mites among the IPS was different. The findings suggest that the intercropped plant species can attract and serve as a reservoir of predatory mites on coffee crops, which could improve the biocontrol of pest mites on coffee.

**Keywords:** conservation biological control; agroecosystem diversification; ecosystem service; herbivores; natural enemies; integrated pest management

## 1. Introduction

Brazil is the largest coffee producer and exporter in the world. The covered area is 1.75 million hectares, from which Minas Gerais has the largest cultivated area (1.22 million hectares); the most cultivated coffee species in the country is *Coffea arabica* L. (Rubiaceae) [1]. Minas Gerais is the state with the largest crop area, corresponding to 71.7% of the occupied area of this species nationwide [1]. Pests and diseases normally cause economic and quality losses on coffee crops [2]. The most frequent strategy in controlling them is the use of chemical inputs such as fungicides, insecticides, and acaricides [3,4]. However, the impact of these pesticides does not only affect the target pest species but also beneficial organisms that play important roles in this agroecosystem due to the ecological services they provide, such as pollination and biocontrol [5–7]. In addition to the impact on biodiversity, the continued use of pesticides also reduces the quality of human life, due to the loss of ecosystem services, increase in food production costs, and health concerns [8–12].

Conservation biological control is defined as the enhancement of natural enemy populations in the agroecosystem, which could be an alternative to conventional practices to control pests in the crop [13–15]. This management can be performed with a variety of integrated strategies using the local biodiversity [16,17]. One strategy is the introduction of intercropped plants that are effective in attracting and favoring key natural enemies without attracting pests in the main crop [18,19]. Among the characteristics used to select

the intercropped plants is the presence of extrafloral nectar and pollen, which are important sources of energy to attract natural enemies and pollinators (i.e., wasps, ants, mites, bees, and chrysopids) [20–27]. Because of their beneficial properties, trees of the genus *Inga* and *Senna* are intercropped with coffee by farmers in some Brazilian regions [28–30]. In addition to providing shade and wood, and fixing nitrogen, these trees also possess extrafloral nectaries that enhance pest control [22,23,31,32]. In addition, *Varronia curassavica* Jacq. is an aromatic perennial shrub that provides pollen during the whole year [33]. Together in the system, they can be a good strategy for conservative biological control by providing alternative food to natural enemies [22,23,28,30,34].

Increasing plant diversity enhances coffee productivity and soil quality, attracts natural enemies, and increases predation and parasitism in this agroecosystem [22,23,35,36]. Some predatory mites, such as phytoseiids from the genus *Amblyseius, Euseius, Neoseiulus, Galendromus*, and *Iphiseiodes*, prey on herbivore mites and also feed on some plant sources, such as pollen and nectar [37–39]. In some studies, predatory mites have been associated with nectar, pollen, and spontaneous plants that grow naturally intercropped with the main crop [20,21,25,40–43]. However, little is known about the effect of intercropped plants with nectaries and pollen on herbivores and predatory mites in coffee crops [25,34,44].

Coffee plants house mites with different food habits, such as predators, mycophagous, herbivores, and other species that do not have a well-known food source reported [4,45–48]. Some of the most common predatory mites found on these plants are from the families Phytoseiidae, Ascidae, Stigmaeidae, Cunaxidae, Cheyletidae, and Anystidae [4,45,47–52], with phytoseiid mites being one of the most well-known and studied [37]. The most economically important herbivore mite in this culture is *Oligonychus ilicis* (McGregor) (Tetranychidae). It causes significant economic losses due to defoliation, premature leaf drop, and reduction in plant photosynthesis [53,54]. The second most important pest is *Brevipalus phoenicis* (Geijskes) (Tenuipalpidae). Although both mites occur throughout the year, dry periods are the most favorable for their development [55]. Additionally, *B. phoenicis* can transmit the coffee ringspot virus, which causes defoliation of the plants [55,56]. This last species comprises several cryptic species [57,58] divided into a complex of eight species. Before this division, *B. phoenicis* was the only species recorded on this crop. Subsequently, there are only records of *B. yothersi* Baker and *B. papayensis* Baker as species belonging to the *B. phoenicis* species complex and being able to transmit the coffee ringspot virus [4,47,48,51,52,58–60]. Additionally, *Polyphagotarsonemus latus* (Banks) (Tarsonemidae) is a pest on coffee that damages the leaves, curling them downward [16,56,61–65]. This species is found in higher densities in rainy seasons [55].

Our aim in this work was to evaluate if the distance from intercropped plants affects the communities of mites present in coffee crops. For this, we used an established intercropped coffee system and evaluated whether the insertion of the intercropped plants providing nectar and pollen modifies (I) the abundance and richness of mites and their feeding behavior, and (II) the dissimilarity of predator and herbivore mites over a gradient of distance from these plants. We also evaluated whether (III) the selected intercropped plants are able to harbor different communities of predator and herbivore mites, considering that the intercropped plants can provide food resources for them [25,44]. We hypothesized that coffee plants closer to the intercropped plants will have more predators, and consequently, fewer pest mites. We also hypothesized that the mite community of these groups tends to become more distinct in coffee plants as the distance from the intercropped plants increases. Finally, we predict that different intercropped plant species may harbor different mite communities, which could contribute to a conservation biological control strategy in coffee systems.

## 2. Materials and Methods

### 2.1. Study Area

The experiment was carried out at the EPAMIG Experimental Farm Station, in Patrocínio county (18°59′52.0″ S 46°58′59.8″ W), Minas Gerais, Brazil, in the Cerrado



biome (Figure 1). The region is located at the mesoregion of Triângulo Mineiro and Alto Paranaíba [66]. The sample area was characterized by a diversified system (1080 m²) with 3 replicated plots in 3 random locations in the planting at least 200 m from each other. The diversified system is surrounded by two rows of three intercropped plant species (IPS): two *Inga edulis* Mart. (Fabaceae) trees, one *Senna macranthera* (Dc. ex collad.) H.S. Irwin and Barnaby (Fabaceae) tree, and six shrubs of *Varronia curassavica* (Cordiaceae) in each row (Figure 1). The amount of each IPS was determined according to the size of the plot and the plants. *Inga edulis* and *S. macranthera* have extrafloral nectaries that produce nectar all day long and year-round, and *V. curassavica* produces flowers during the whole year [31–33]. No pesticides were applied in the study plots. Fertilization was made with chemical fertilizers and the spontaneous plant growth was controlled by a brush cutter.

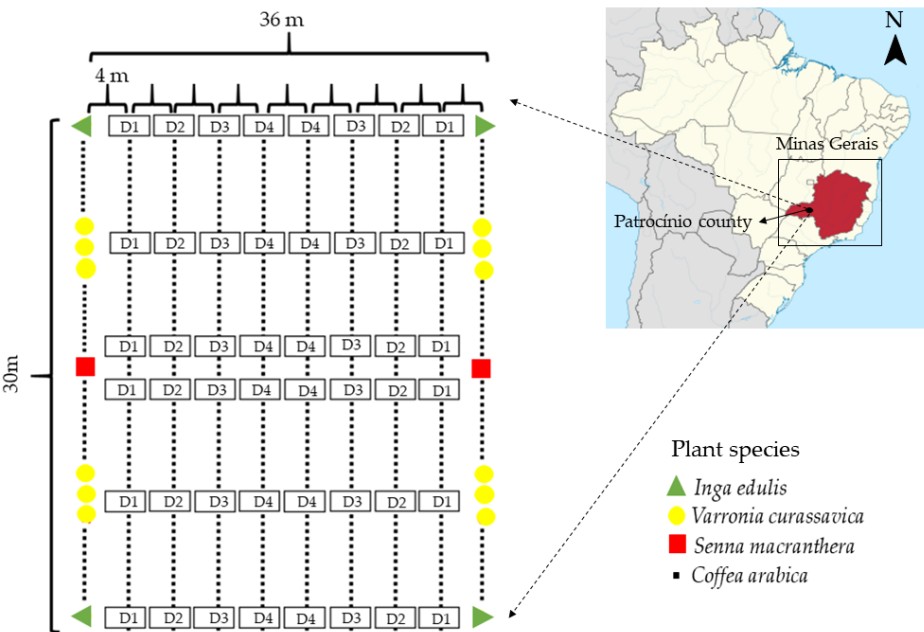

**Figure 1.** Map showing the spatial arrangement of the sampling plot, diversified coffee system (1080 m²), at Patrocínio county, Minas Gerais, Brazil. Green triangles represent *Inga edulis*, yellow circles represent *Varronia curassavica*, red squares represent *Senna macranthera*, and black squares represent coffee plants. D1, D2, D3, and D4 represent the coffee plants sampled in their respective distances from the IPS: D1 represents distance 1 from the IPS (4 m); D2 represents distance 2 from the IPS (8 m); D3 represents distance 3 from the IPS (12 m); and D4 represents distance 4 from the IPS (16 m). Figure adapted from [24].

*2.2. Data Collection*

To evaluate the effect of intercropped plants on mite communities along a distance gradient in coffee crops, we sampled coffee plants four, eight, twelve, and sixteen meters from the intercropped plants (Figure 1). In each plant, we detached four leaves, beginning with the third leaf pair from the distal end of the shoot; this was conducted for two shoots on each plant. Two branches in the middle third of each bush facing both sides of the coffee rows were sampled (adapted [67]). A total of four plants were sampled every four meters, which corresponded to consecutive coffee rows from each IPS (Figure 1). A total of 192 leaves were sampled per plot totaling 576 leaves in this experiment. Because the mite species repeated themselves at each time point, the fourth and fifth sampling followed the same procedure but instead of four leaves per plant, just 25% of the material was sampled, which means one leaf per plant. A total of 48 leaves/plot was sampled, totaling 144 leaves (adapted from Ref. [45]). To sample the IPS, five leaves of each plant species were detached randomly per row for a total of 10 leaves per plant species/plot, totaling 90 leaves. The material was collected at five times points: June 2020 and September 2020 (dry season), and

October 2020, November 2020, and February 2021 (rainy seasons). The samples were placed separately in paper bags and kept at low temperatures with Gelo-X® inside a cooler box until they arrived at the laboratory of Acarology at the Department of Entomology at the Federal University of Viçosa, Minas Gerais (20°45′37.0″ S 42°52′07.0″ W). At the laboratory, samples were kept in a refrigerator at approximately 10 °C for a period of up to 10 days. The leaves were examined with a stereoscopic microscope and all mites were mounted on glass slides in Hoyer's medium for identification [68]. Morphological identification was made on a microscope with phase contrast using a dichotomous key [58,69–76].

Different amounts of leaves were sampled at different time points because only 25% of the leaves were sampled in the fourth and fifth sampling. So, to maintain a consistent identification pattern, from the first to third sampling, 25% of each slide was identified and in the fourth and fifth all the slides were identified. To accomplish this, the cover slide was divided into four identical quadrants, each one corresponding to 25% of the slide. The identified quadrant was raffled. If in the raffled quadrant there were no mites, the identified was the anterior one.

Females, males, and immature mites were included to analyze abundance. To richness, only females were included because it was not possible to identify or morphotype males. When assessing Eriophyoidea mites, some individuals were mounted on microscope slides with modified Berlese's medium [77] for morphotyping, and some individuals were preserved in 70% ethanol for posterior identification. To measure the abundance, all Eriophyoidea from the first sample were counted. From the second up to the fifth sample, we used the mean number of mites per leaf. Hence, all individuals of the same morphospecies were counted in a 1 cm$^2$ quadrant in the abaxial surface of each leaf in the region near to petiole and midrib. So, the leaves were measured and the number of specimens per leaf was estimated. Only for this superfamily were all specimens included in richness.

All specimens were deposited in the mite reference collection of the Laboratory of Acarology at the Department of Entomology at the Federal University of Viçosa, Minas Gerais, Brazil.

### 2.3. Statistical Analyses

We tested the effect of the distance gradient on the abundance of the herbivorous and predatory mites using generalized linear mixed models (GLMMs) with a Poisson error structure (and a log-link function) with the "lme4" v.3.1-147 package [78]. In the full model for each response variable, we specified a fixed polynomial effect of the distance gradient (i.e., including a quadratic term in the regression function) to account for potential nonuniformity in response. Each model included a random intercept for "plot" (*n* = 3) to account for the non-independence of the samples located within each plot. We used the Gaussian distribution, and no overdispersion was found for the best-fitting models. For the construction of the full model for the abundance of mites, we carried out a model simplification process using the "AICcmodavg" package [79]. We determined the minimum adequate model(s) by comparing akaike information criterion corrected (AICc) values and AICc weights (AICcWt) for sub-models consisting of (1) a polynomial DG * behavior model, (2) a polynomial DG + behavior model, (3) a linear DG * behavior model, (4) a linear DG + behavior model, or (5) a null intercept-only model. Models within 2 ΔAICc units of the top model (i.e., the model with the lowest AICc and highest AICcWt values) were considered to have equivalent explanatory power [80]. For the final model, we used the approach of Nakagawa and Schielzeth [81] to estimate absolute model fit using marginal R2 GLMM (variance explained by just the fixed effects) and conditional R2 GLMM (variance explained by both fixed and random effects). To evaluate the difference of abundance among gradient distance and/or food behavior, we used the Analysis of Deviance e (Type III) performed with package "car" version 3-1.0 [82] and Contrast Analysis with package "lsmeans" version 2.30-0 [83].

To investigate changes in the composition of mite communities along the distance gradient, a species dissimilarity matrix based on the Bray–Curtis distance metric was calculated among the plots [84]. From this, we calculated the average pairwise dissimilarity of each sample to the centroid of the first distance of the coffee plant sampled using the function "vegdist" in the "vegan" v.5-5 package [85]. This response measure, therefore, reflects "dissimilarity to distance of IPS" in terms of species composition. We tested the effect of the distance gradient on mite "dissimilarity to distance of IPS" using separate linear models (LMs) with a gaussian error structure (and an identity link function). We specified a fixed polynomial effect of the distance gradient and, as described above, we carried out model simplification using AIC comparisons of polynomial, linear, and null sub-models. Finally, we estimated the coefficient of determination of final models using the "rsq" v.1.1 package [86].

We tested the variation in the composition of predators and herbivores among plants with the permutational analysis of multivariate dispersions (PERMDISP) and the permutational multivariate analysis of variance (PERMANOVA) using the function *betadisper* and *adonis*, respectively, both from the "vegan" package version 2.4-2 [87]. When we found differences in composition among treatments, we performed a pairwise comparison using the pairwise adonis function with a Bonferroni correction [88]. All analyses were performed in R version 3.5.1 [89].

## 3. Results

### 3.1. Abundance and Richness of Mites

In total, we collected 8946 specimens belonging to 13 families and 37 species, divided into 4 feeding behaviors: generalist, mycophagous, herbivores, and predators. Out of the thirty-seven species found, three were generalists (Tydeidae), five were mycophagous (Acaridae, Glycyphagidae, Tarsonemidae, and Winterschmidtiidae), twelve were herbivores (Eriophyidae, Diptilomiopidae, Tarsonemidae, Tenuipalpidae, and Tetranychidae), and sixteen were predators (Iolinidae, Phytoseiidae, and Stigmaeidae). On coffee, Tenuipalpidae was the most abundant family ($n$ = 656) but only one species was reported, followed by Tydeidae ($n$ = 126) with three species and Tetranychidae ($n$ = 96) with two species found. On the IPS, Eriophyidae was the most abundant family recorded ($n$ = 6252) with four species, followed by Phytoseiidae ($n$ = 282) with twelve species and Tenuipalpidae ($n$ = 257) with one species (Tables 1 and 2).

The best model that described the mite abundance was the polynomial with predictors *behavior + distance* ($R^2$(m) = 0.74, $R^2$ (C) = 0.74, AICc = 542.01 and AICcWt = 0.89) (Table S1). We found an abundance of mites differed among their feeding behavior ($X^2$ = 68.637, df = 3, $p < 0.001$) and along the gradient distance from intercropped plants ($X^2$ = 104.825, df = 2, $p < 0.001$, Figure 2). Regarding the difference in abundance among the feeding behavior along the distance gradient, the abundance of herbivores was higher than the other feeding behaviors ($p < 0.0001$). Additionally, no variation among predators, mycophagous, and generalist mites were observed ($p > 0.05$).

### 3.2. Dissimilarity of Mites

The dissimilarity of species along the distance gradient for both herbivorous and predatory mites presented the polynomial model as the most adjusted (see Table S1). We found that the dissimilarity of herbivores and predators increased over the distance gradient (F = 22.534, $p < 0.0001$, $R^2$ = 0.63, Figure 3a and F = 44.464, $p < 0.0001$, $R^2$ = 0.80, Figure 3b, respectively).

### 3.3. Communities of Herbivorous and Predatory Mites on the IPS

The composition of herbivorous mites on IPS was different (PERMDISP: F = 0.21, $p$ = 0.80; PERMANOVA: $R^2$ = 0.22, $p < 0.05$, Figure 4a). *Inga edulis* was different from *S. macranthera* ($R^2$ = 0.32, $p$ = 0.001) and *V. curassavica* ($R^2$ = 0.32, $p$ = 0.002), but *V. curassavica* and *S. macranthera* were not different ($R^2$ = 0.09, $p$ = 0.36).

**Table 1.** Richness and composition of mite species in the intercropped coffee systems between June 2020 and February 2021: *Vc Varronia curassavica*, *Sm Senna macranthera*, *Ie Inga edulis*, *Ca1 Coffea arabica* row 1, *Ca2 C. arabica* row 2, *Ca3 C. arabica* row 3, and *Ca4 C. arabica* row 4.

| Feeding Behavior | Mite Family | Mite Species | Plant Species | | | | | | |
|---|---|---|---|---|---|---|---|---|---|
| | | | *Vc* | *Sm* | *Ie* | *Ca1* | *Ca2* | *Ca3* | *Ca4* |
| **Generalists** | Tydeidae | *Lorryia formosa* Cooreman | 1 | 1 | 48 | 1 | | 1 | |
| | Tydeidae | *Lorryia* sp1 | 3 | 34 | 8 | 14 | 7 | 22 | 10 |
| | Tydeidae | *Lorryia* sp2 | | | 5 | 1 | | | 1 |
| **Mycophagous** | Acaridae | *Tyrophagus putrescentiae* (Schrank) | 26 | 27 | 84 | 7 | 10 | 12 | 16 |
| | Astigmatina * | Astigmatina sp. | 3 | | | | | | |
| | Glycyphagidae | *Lepidoglyphus destructor* (Schrank) | | | 1 | | | | 1 |
| | Tarsonemidae | *Daidalotarsonemus savanicus* Rezende, Lofego and Ochoa | | | 4 | 1 | | | |
| | Tarsonemidae | *Tarsonemus confusus* Ewing | | | 7 | 6 | 11 | 1 | 2 |
| | Winterschmidtiidae | Winterschmidtiidae sp. | 4 | | 1 | | | | |
| **Herbivores** | Eriophyidae | *Aculus* sp. | | 1200 | | | | | |
| | Eriophyidae | Eriophyidae sp1 | | | 4885 | | | | |
| | Eriophyidae | Eriophyidae sp2 | 70 | | | | | | |
| | Eriophyidae | Eriophyidae sp3 | 97 | | | | | | |
| | Diptilomiopidae | Diptilomiopidae sp. | 18 | | | | | | |
| | Tarsonemidae | *Polyphagotarsonemus latus* (Banks) | 65 | | | | | | |
| | Tenuipalpidae | *Brevipalpus yothersi* Baker | 15 | 144 | 77 | 95 | 152 | 110 | 100 |
| | Tetranychidae | *Atrichoproctus uncinatus* Flechtmann | 12 | 7 | 12 | | | | |
| | Tetranychidae | *Mononychellus planki* (McGregor) | | 3 | | | | | |
| | Tetranychidae | *Oligonychus coffeae* (Nietner) | | | | 7 | 4 | 12 | 4 |
| | Tetranychidae | *Oligonychus ilicis* (McGregor) | | | 1 | 1 | 1 | 4 | 2 |
| | Tetranychidae | *Tetranychus* sp. | | | 2 | | | | |
| **Predators** | Iolinidae | *Pseudopronematulus nadirae* Silva, Da-Costa and Ferla | 41 | 61 | 1 | 5 | 7 | 10 | 4 |
| | Iolinidae | *Pausia* sp. | 5 | 7 | | | | | |
| | Phytoseiidae | *Amblyseius* aff. *chiapensis* | 1 | | 2 | | | | |
| | Phytoseiidae | *Amblyseius* aff. *impressus* | | | | 1 | | | |
| | Phytoseiidae | *Amblyseius* sp. | | | 1 | | | | |
| | Phytoseiidae | *Euseius alatus* DeLeon | | | 4 | | | | |
| | Phytoseiidae | *Euseius citrifolius* (Denmark and Muma) | 5 | 6 | 7 | 4 | 2 | 2 | 1 |
| | Phytoseiidae | *Euseius concordis* (Chant) | 1 | 1 | 11 | 4 | 5 | 3 | 3 |
| | Phytoseiidae | *Euseius sibelius* (De Leon) | 1 | 6 | 1 | | | | |
| | Phytoseiidae | *Iphiseiodes zuluagai* Denmark and Muma | | | 32 | | | 2 | 2 |
| | Phytoseiidae | *Galendromus annectens* (De Leon) | | 4 | 2 | | | | |
| | Phytoseiidae | *Neoseiulus tunus* (De Leon) | 6 | | | | | | |
| | Phytoseiidae | *Typhlodromalus aripo* DeLeon | 4 | 4 | 1 | | | | |
| | Phytoseiidae | *Typhlodromips mangleae* De Leon | | 1 | 4 | | | | |
| | Stigmaeidae | *Agistemus floridanus* Gonzalez | 38 | 13 | 78 | 1 | | 1 | 1 |
| | Stigmaeidae | *Agistemus brasiliensis* Matioli, Ueckermann and Oliveira | | 2 | | | | | |

* Cohort.

**Table 2.** Abundances of mite species in the intercropped coffee systems between June 2020 and February 2021: **Vc** *Varronia curassavica*, **Sm** *Senna macranthera*, **Ie** *Inga edulis*, **Ca1** *Coffea arabica* row 1, **Ca2** *C. arabica* row 2, **Ca3** *C. arabica* row 3, and **Ca4** *C. arabica* row 4.

| Feeding Behavior | Mite Family | Plant Species | | | | | | |
|---|---|---|---|---|---|---|---|---|
| | | Vc | Sm | Ie | Ca1 | Ca2 | Ca3 | Ca4 |
| **Generalists** | Tydeidae | 6 | 37 | 76 | 35 | 19 | 50 | 22 |
| **Mycophagous** | Acaridae | 32 | 37 | 109 | 19 | 17 | 19 | 25 |
| | Astigmatina * | 3 | | | | | | |
| | Glycyphagidae | | | 1 | | | | 1 |
| | Tarsonemidae | 111 | 14 | 23 | 7 | 12 | 3 | 4 |
| | Winterschmidtiidae | 4 | | 1 | | | | |
| **Herbivores** | Eriophyidae | 167 | 1200 | 4885 | | | | |
| | Diptilomiopidae | 18 | | | | | | |
| | Tarsonemidae | 108 | | | | | | |
| | Tenuipalpidae | 15 | 158 | 84 | 130 | 208 | 171 | 147 |
| | Tetranychidae | 19 | 14 | 18 | 16 | 14 | 27 | 39 |
| **Predators** | Iolinidae | 85 | 134 | 2 | 8 | 12 | 20 | 11 |
| | Phytoseiidae | 37 | 53 | 192 | 16 | 15 | 15 | 21 |
| | Stigmaeidae | 48 | 20 | 129 | 1 | | 1 | 1 |

\* Cohort.

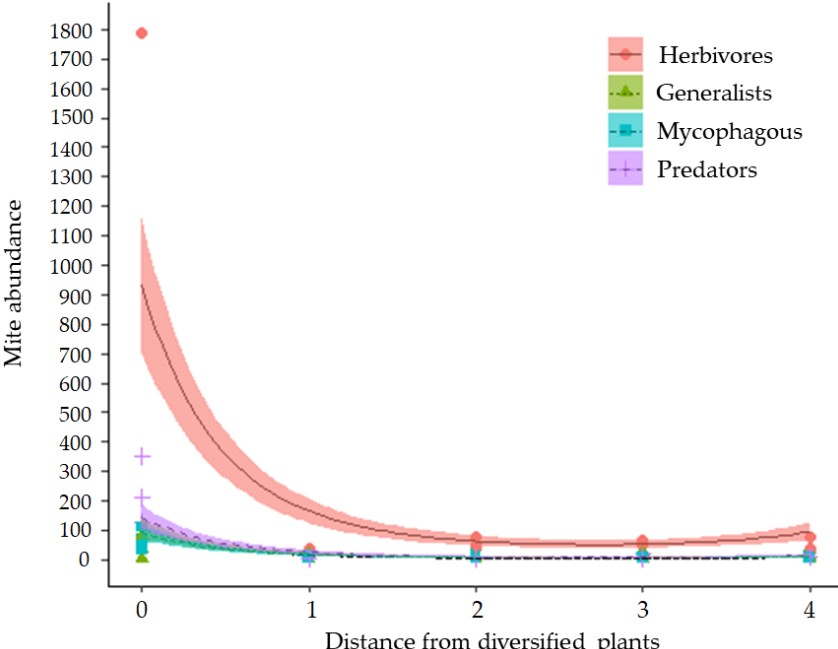

**Figure 2.** Mite abundance of different feeding behaviors along a distance gradient. Distance 0 refers to mites on the IPS and distance 1 to 4 refers to the mites on coffee plants at their respective distances.

The composition of predators on IPS was also different (PERMDISP: F = 0.18, *p* = 0.83; PERMANOVA: R$^2$ = 0.33, *p* < 0.001, Figure 4b). *Inga edulis* was different from *V. curassavica* (R$^2$ = 0.32, *p* = 0.002) and *S. macranthera* (R$^2$ = 0.32, *p* = 0.003), but *V. curassavica* and *S. macranthera* were not different (R$^2$ = 0.09, *p* = 0.37).

The predatory mite species found in all IPS were *Pseudopronematulus nadirae* Silva, DaCosta, and Ferla, *Euseius citrifolius* (Denmark and Muma), *E. concordis* (Chant), *E. sibelius* (De Leon), *Typhlodromalus aripo* DeLeon, and *Agistemus floridanus* Gonzalez. In general, *I. edulis* houses more predator species than the other IPS, with *Amblyseius* sp., *E. alatus* DeLeon, *Iphiseiodes zuluagai* Denmark and Muma, *Agistemus brasiliensis* Matioli, Ueckermann, and Oliveira occurring only on this plant species. *Varronia curassavica* has only one exclusive species (*Neoseiulus tunus* (De Leon)) and *S. macranthera* did not have any exclusive species.

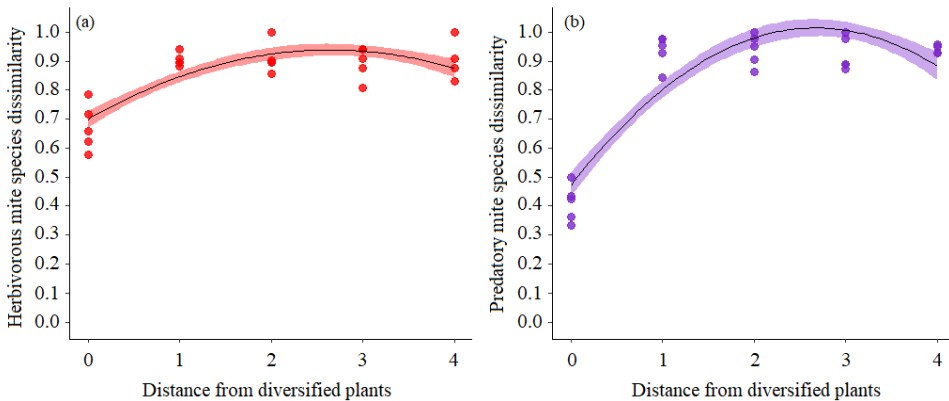

**Figure 3.** Variation in (**a**) herbivorous mite community dissimilarity and (**b**) predatory mite community dissimilarity from the IPS. Distance 0 refers to mites on the IPS and distance 1 to 4 refers to the mites on coffee plants at their respective distances.

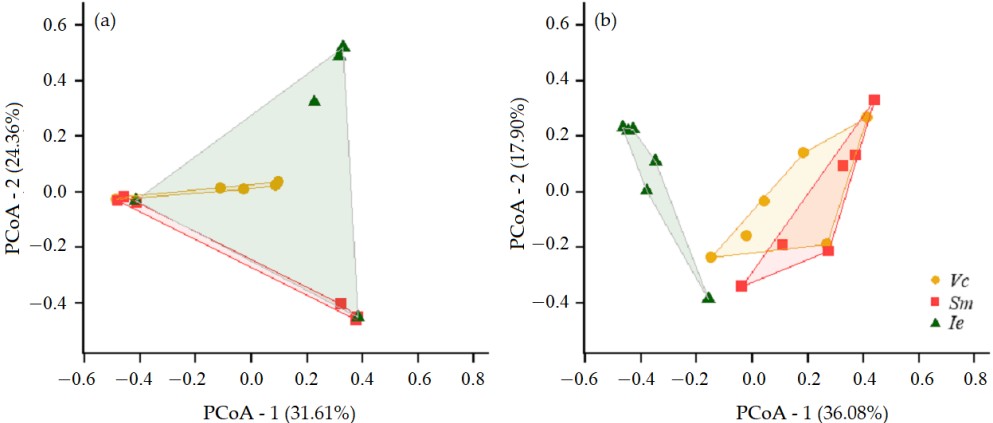

**Figure 4.** Principal coordinate analysis (PCoA) plot based on the Bray–Curtis dissimilarity index from samples of predatory and herbivorous mites composition between the IPS. Composition of herbivores (**a**) and predators (**b**) from the IPS. Yellow circles represent *Varronia curassavica* (*Vc*), red squares represent *Senna macranthera* (*Sm*), and green triangles represent *Inga edulis* (*Ie*).

*Brevipalpus yothersi* and *Atrichoproctus uncinatus* Flechtmann were the only herbivore species common in all IPS. Eriophyidae sp1., *O. ilicis* and *Tetranychus* sp. are the species exclusive to *I. edulis*. Eriophyidae sp2 and sp3, Diptilomiopidae sp., and *P. latus* were the species that occur only on *V. curassavica*, and *Aculus* sp. and *Mononychellus planki* (McGregor) were the only two species exclusive from *S. macranthera*. The number of Eriophyidae on *I. edulis* was high (4885), despite having only one eriophyid species in this plant, while *V. curassavica* and *S. macranthera* had different species in a much lower number. It is important to notice that despite being a pest on coffee, *P. latus* only occurred on *V. curassavica* with no records on coffee.

## 4. Discussion

The data presented here show that the intercropped plants are a reservoir of natural enemies, as they harbor several species of predators. The abundance of mites differed in their feeding behavior and distance with the dissimilarity of predators and herbivores increasing along a gradient of distance.

Diversified coffee production systems attract predators, mostly wasps and parasitoids, increase biological control, improve soil quality, and decrease spontaneous plants [22,23,25,35,36,44,90]. Studies with *Inga* spp. demonstrate the positive effects of this plant intercropped with coffee, increasing natural control of coffee berry borers, coffee leaf miners, and important coffee pests by attracting wasps and parasitoids due to the

provision of nectar, and consequently enhancing the coffee yield [22,23]. So, in this work, we observed that not just *I. edulis*, but also *V. curassavica* and *S. macranthera* house predators. All predatory mites found on coffee were also found on the IPS (except for *A.* aff *impressus* that has only one individual recorded), meaning that coffee plants are benefiting from the IPS harboring predators that could control their pest mites. In addition to providing additional food sources, such as pollen and nectar, the intercropped plants also can provide shelter, a favorable microclimate, and a habitat where prey is present [91].

The composition of herbivorous and predatory mites was different in *I. edulis* and similar in *V. curassavica* and *S. macranthera*. *Inga* plants harbored many more predators than other IPS, did not harbor *P. latus*, and only harbored a few *O. ilicis* (*n* = 1), two herbivore pests of economic impact on coffee crops. *Senna macranthera* did not house any of these pest species but also harbored some predators. This is an important finding because these plants have the potential of harboring different natural enemies and do not attract coffee pest mites. Thus, due to the differences in natural enemies' species in each IPS, the use of the three plants together in coffee crop systems stimulated the diversity of predators. The natural enemies found in this work belong to the families Stigmaeidae, Iolinidae, and Phytoseiidae, with the latter having the greatest number of species. Most of the species found here are known to feed and reproduce on a wide range of prey and Phytoseiidae, for example, also feed on pollen and nectar that can constitute an important part of their diet [37,38,92–96]. Phytoseiidae is an important predatory family that is extensively studied and used for the biological control of pests [97]. Species of the genus *Amblyseius*, *Neoseiulus*, *Galendromus*, and *Typhlodromalus* have records of feeding on pollen, but also on Eriophyidae, Tarsonemidae, Tetranychidae, Tydeoidea, and other herbivore families [37,38]. Additionally, species of the genus *Euseius* and *Iphiseiodes* are pollen feeders and are also generalists that prey on a range of mites, and *B. phoenicis* is one of them [38,98–100]. Additionally, *I. zuluagai* can feed on sugary substances and it is able to be reared on a range of alternative food sources [100,101]. At the time of the studies cited above, *B. phoenicis* was not divided into a species complex yet [58], and all the species of this complex were called *B. phoenicis*. So, we cannot be certain which species of these complexes were recorded in those studies.

The feeding habits of Tydeoidea are diverse [102], and Iolinidae are associated with several herbivorous species, preying on Eriophyidae, Tenuipalpidae, and Tetranychidae [92,94–96]. Other mites that feed on Eriophyoidea are Stigmaeidae, which can also feed on Tetranychidae [93,103]. In this work, there were a great amount of Eriophyidae and other herbivorous species belonging to Diptilomiopidae, Tenuipalpidae, Tarsonemidae, and Tetranychidae. Thus, the larger number of herbivores in the system is mainly due to the high number of eriophyids found on the IPS (Tables 1 and 2). Additionally, referring to mycophagous mites, there are some species that belong to the family Acaridae, Glycyphagidae, Winterschmidtiidae, and Tarsonemidae, which in coffee may be associated with the presence of domatia, influencing the distribution and abundance of the group [104]. Knowing that the pests on coffee vary in quantity during the year, the mites on the IPS and also those on coffee plants can probably be preyed on and serve as alternative food by the predators recorded here, mainly when the pests on coffee are in a low population [45,55,105]. Furthermore, the IPS stimulates the diversity of natural enemies of other insect pests in this crop [22,23,106].

Thus, these results suggest that the IPS attracts predatory and herbivorous mites that are not pests on coffee. These herbivores could be alternative prey to increase the population of predators on these plants that could later migrate to coffee plants and control their pest mites. Once this happens, it could enhance the conservation of biological control in the area. Understanding the dynamics of predators and prey in this system would facilitate management practices and favor the establishment of sustainable pest mite control on coffee crops.

**Supplementary Materials:** The following supporting information can be downloaded at https://www.mdpi.com/article/10.3390/agriculture13020285/s1, Table S1: Comparison of the likelihood of model fit for null, linear, and polynomial models of the response variable to the distance gradient (DG) and behavior. AICc: akaike information criterion corrected; ΔAICc: difference in AICc between

the model and the model with the smallest AICc; AICcWt: model weight according to ΔAICc. Values in bold represent the best models.

**Author Contributions:** Conceptualization, J.J.F.; methodology, J.J.F., P.H.M.G.N., M.O.K., S.D.P., A.C.C. and E.F.M.; formal analysis, G.J.d.A.; investigation, J.J.F.; resources, A.P. and M.V.; data curation, J.J.F., P.H.M.G.N., M.O.K., S.D.P., A.C.C. and E.F.M.; writing—original draft preparation, J.J.F.; writing—review and editing, G.J.d.A., N.J.F., A.P. and M.V.; visualization, J.J.F.; supervision, G.J.d.A., N.J.F., A.P. and M.V.; project administration, A.P. and M.V.; funding acquisition, A.P. and M.V. All authors have read and agreed to the published version of the manuscript.

**Funding:** This research was funded by "Conselho Nacional de Desenvolvimento Científico e Tecnológico" (CNPq, 312784/2021-0), "Fundação de Amparo à Pesquisa de Minas Gerais" (FAPEMIG, CAG-PPM-00270/18), "Coordenação de Aperfeiçoamento de Pessoal de Nível Superior—Brasil" (CAPES)—Finance Code 001, and "Consórcio Brasileiro de Pesquisa e Desenvolvimento do Café" (CBP&D-Café, 10.18.20.049.00.04).

**Institutional Review Board Statement:** Not applicable.

**Data Availability Statement:** Not applicable.

**Acknowledgments:** The authors are grateful to Enrico de Lillo, Marcello De Giosa, Tairis Da-Costa, and Gabriel Lima Bizarro for taxonomic discussion and help. We also thank the colleagues from the Laboratory of Acarology at the Federal University of Viçosa for assistance with practical work, suggestions, and discussions.

**Conflicts of Interest:** The authors declare no conflict of interest. The funders had no role in the design of the study; in the collection, analyses, or interpretation of data; in the writing of the manuscript; or in the decision to publish the results.

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
