# Peer review of "Intercropped Plants Provide a Reservoir of Predatory Mites in Coffee Crop"

_agriculture, doi:10.3390/agriculture13020285_

Round 1

Reviewer 1 Report

Agriculture-2116604
Reviewer comments

Intercropped plants as a reservoir of predatory mites in coffee crop 

This work evaluated the effect of three different “Intercropped plant species” (Inga edulis, Senna macranthera, and Varronia curassavica) in a coffee planting system on the mite populations, both on the IPS and coffee plants at different distances from the IPS. Results demonstrate that these plants affect the predatory and herbaceous mites differentially and may be effective as a conservation biological pest control. The research presented in this manuscript is novel and falls within the scope of this journal. The work was conducted appropriately for rigorous evaluation of the treatments. There are important, exciting potential consequences for pest management in commercial coffee production, including adoption of a conservation biological method and reduced use of pesticides.

I recommend this manuscript be accepted with revisions. There are a few minor edits, including tense agreements, grammatical changes, clarified sentences, and punctuation corrections. A few minor clarifications are needed for the Methods. There is a bit more analysis that may be appropriate. My suggestions are outlined below and as track changes in the PDF document.

In the Introduction, could you please add something to the introduction for readers who are unfamiliar with:

- Coffee production systems. A very brief description of conventional layouts and spacing, planting systems, major mite pests, commercial standard control methods). Is the system in this work the same as a conventional system aside from the IPS (i.e. does the work translate directly or would modifications be needed)?

- Mite behavior in coffee systems. For example: Are mites typically problematic throughout the season or only certain times of the year? What environmental factors contribute? How prevalent are they and approximately what amount of economic damage do they cause? Are they exacerbated by other practices? Are they typically present throughout the field or only in spots? How quickly do they move within and between blocks? What time of the season are they most active. How does this relate to the sampling time points?

In the Methods, a bit more clarification is needed. It seems the sample collection took place at multiple timepoints, but this is not stated explicitly early enough in the methods. Please clarify. I would also suggest using more consistent language throughout the manuscript to refer to components of the research design. For example, could ‘rows’ be used  instead of ‘lines’? Could ‘plots’ be used consistently instead of ‘areas’ or ‘replicas’?

Was leaf damage rated in this trial? It would be the best indication in this or future work for the impact of IPS as biocontrol.

Figure 2. Does this represent the entire planting, or just one plot (if the latter, could one plot be indicated in this diagram)? Are there additional rows in between the four distances that were evaluated or are rows spaced 4m apart? Please clarify. 

In the Analysis, should the feeding behavior along distance gradient be calculated for the different IPS separately? It seems these plants hosted very different mite populations and therefore may have impacted this gradient differently. A better discussion of the ramifications of these differences for biocontrol would be justified. Are there other IPS that may be appropriate based on this work?

In addition, in the Analysis, variation in mite communities at distances from the IPS included distances ‘0’ through ‘4’ lines/rows of coffee plants (Figure 4). Should these correlation analyses include distance ‘0’? I would suggest also doing calculations beginning at distance 1, omitting distance 0 (the IPS). This is because the IPS are different species of plant than coffee, therefore the difference in communities may indicate that the different host plants harbor different mite communities, and not necessarily that the IPS influence mite communities in adjacent coffee plants. Including only coffee may be a better representation of how mite populations are affected in one species, i.e. the target crop. If there is no difference in this analysis, it would indicate that the mite population in the crop was not really affected, and meaningfully, pest pressure was not reduced due to the inclusion of IPS (or vice versa).

Overall, this manuscript includes excellent work, which was well conducted, clearly and adequately explained, and impactful to the industry.

Author Response

This work evaluated the effect of three different “Intercropped plant species” (Inga edulis, Senna macranthera, and Varronia curassavica) in a coffee planting system on the mite populations, both on the IPS and coffee plants at different distances from the IPS. Results demonstrate that these plants affect the predatory and herbaceous mites differentially and may be effective as a conservation biological pest control. The research presented in this manuscript is novel and falls within the scope of this journal. The work was conducted appropriately for rigorous evaluation of the treatments. There are important, exciting potential consequences for pest management in commercial coffee production, including adoption of a conservation biological method and reduced use of pesticides.

I recommend this manuscript be accepted with revisions. There are a few minor edits, including tense agreements, grammatical changes, clarified sentences, and punctuation corrections. A few minor clarifications are needed for the Methods. There is a bit more analysis that may be appropriate. My suggestions are outlined below and as track changes in the PDF document.

Reply: Done. We have reviewed and accepted the suggestions you made in the pdf document.

You might consider displaying this data with cells colored with increasing intensity for higher values as a more effective visual representation. You may also including % values of the number of mite species in parenthesis. Neither are necessary, just suggestions.

Reply:  Thanks for this suggestion. We discussed it and based on other works of Araújo (Araújo et al. 2021: https://doi.org/10.1007/s00442-021-04920-z) we chose to keep the original version.

In the Introduction, could you please add something to the introduction for readers who are unfamiliar with:

- Coffee production systems. A very brief description of conventional layouts and spacing, planting systems, major mite pests, commercial standard control methods). Is the system in this work the same as a conventional system aside from the IPS (i.e. does the work translate directly or would modifications be needed)? / Mite behavior in coffee systems. For example: Are mites typically problematic throughout the season or only certain times of the year? What environmental factors contribute? How prevalent are they and approximately what amount of economic damage do they cause? Are they exacerbated by other practices? Are they typically present throughout the field or only in spots? How quickly do they move within and between blocks? What time of the season are they most active. How does this relate to the sampling time points?

Reply: Some of these topics were added in the introduction (for instance line 77 - 79; 86).

However, some of them were not the object of this study or are still unknown (for instance: speed/movement between plots; specific details of the mites in each season; economic damage in some cases) and for these reasons we did not add it.

In addition, mite behavior varies from species to species. In one hand, this makes difficult to talk about them in a general way, for example: mites belonging to the tetranychid family are mostly found in spots, while other mite families are found throughout the field. On the other hand, if we added punctual characteristics between different species the text could be extensive and a little confusing. So, these are some of the reasons we decided not to add these general topics in the text.

In the Methods, a bit more clarification is needed. It seems the sample collection took place at multiple timepoints, but this is not stated explicitly early enough in the methods. Please clarify.

Reply: Done (Line 147 - 149)

 I would also suggest using more consistent language throughout the manuscript to refer to components of the research design. For example, could ‘rows’ be used  instead of ‘lines’? Could ‘plots’ be used consistently instead of ‘areas’ or ‘replicas’? 

Reply: Done. We replaced all the suggested terms during the manuscript.

Was leaf damage rated in this trial? It would be the best indication in this or future work for the impact of IPS as biocontrol.

Reply: We understand the importance of evaluating the damage caused by mites in coffee plants for biological control, but leaf damage was not rated in the trial.

Figure 2. Does this represent the entire planting, or just one plot (if the latter, could one plot be indicated in this diagram)? Are there additional rows in between the four distances that were evaluated or are rows spaced 4m apart? Please clarify. 

Reply: Done. This represents a plot of a diversified coffee system. All plants presented in each plot is drawn in the diagram, with no other rows between distances. We modified figure 1 and the legend for a better understanding of the system.

In the Analysis, should the feeding behavior along distance gradient be calculated for the different IPS separately? It seems these plants hosted very different mite populations and therefore may have impacted this gradient differently. A better discussion of the ramifications of these differences for biocontrol would be justified. Are there other IPS that may be appropriate based on this work?

Reply: We appreciate the comment. Our results bring information about the contribution of each intercropped plant in the composition of the mite community present in the IPS. Also, our objective in the present work is to evaluate how the diversification of coffee cultivation can be benefited by its intercropping with IPS with different potential characteristics that attract natural enemies of coffee pests. Evaluating the contribution of each intercropped plant separately would limit the objective of our present study.

In addition, in the Analysis, variation in mite communities at distances from the IPS included distances ‘0’ through ‘4’ lines/rows of coffee plants (Figure 4). Should these correlation analyses include distance ‘0’? I would suggest also doing calculations beginning at distance 1, omitting distance 0 (the IPS). This is because the IPS are different species of plant than coffee, therefore the difference in communities may indicate that the different host plants harbor different mite communities, and not necessarily that the IPS influence mite communities in adjacent coffee plants. Including only coffee may be a better representation of how mite populations are affected in one species, i.e. the target crop. If there is no difference in this analysis, it would indicate that the mite population in the crop was not really affected, and meaningfully, pest pressure was not reduced due to the inclusion of IPS (or vice versa). Overall, this manuscript includes excellent work, which was well conducted, clearly and adequately explained, and impactful to the industry.

Reply: In our study, line 0 presents the starting point (control). Including IPS information in the analyzes supports us in informing whether intercropped plants have the potential to harbor a mite community that has beneficial effects on coffee plants and the radius that this effect can reach within the system. Using the first coffee line (instead of the control) as a starting point can lead to misinterpretations that IPS do not bring beneficial contributions to the control of herbivore mites in the coffee system. This study is a first step towards understanding the potential of the intercropped plants used here. Based on this work, future studies will be carried out focusing on quantitative aspects of these IPS within crops.

Reviewer 2 Report

Thank you for the invitation to fulfill your manuscript titled:

Intercropped plants as a reservoir of predatory mites in coffee crop

The study is very interesting and enriches knowledge, especially in the field of acarology. This manuscript is generally well-written, concise, and clear. Statistical analysis is well chosen and described. The author performed a Permanova analysis, which is the right approach to the methodological assumptions. The results are clearly presented. Material and methods were used accordingly. Please consider the following fixes:

Figures 1, 2, 3, 4, and 5: To improve the resolution, because the quality of the graphics is bad,

Please include the Specific Test Probability Level Score (p =..) if possible.

but do not refer to the result as the assumed level of 0.05.

The correction only applies when p > 0.05 and P > 0.0001.

When the result is P 0.0001, as specified in line 234, this does not apply.

The same is true for upgrade lines 241-247.

Author Response

The study is very interesting and enriches knowledge, especially in the field of acarology. This manuscript is generally well-written, concise, and clear. Statistical analysis is well chosen and described. The author performed a Permanova analysis, which is the right approach to the methodological assumptions. The results are clearly presented. Material and methods were used accordingly. Please consider the following fixes:

Figures 1, 2, 3, 4, and 5: To improve the resolution, because the quality of the graphics is bad.

Reply: Done. We appreciate the comments, and we improved the resolution of all images.

Please include the Specific Test Probability Level Score (p =..) if possible. but do not refer to the result as the assumed level of 0.05. The correction only applies when p > 0.05 and P > 0.0001. When the result is P 0.0001, as specified in line 234, this does not apply. The same is true for upgrade lines 241-247.

Reply: Done. Whenever possible, we used p values, but sometimes this was not possible due to low p values (when P<0.001)

Reviewer 3 Report

Comments to the Authors

- The data and interpretations, however, could have been more informative if the experiments had been repeated for at least one more year to improve understanding and consistency of the findings. Therefore, I think that the data collected is moderately sufficient to support the main hypothesis of the authors. Related to this, considering that the study was carried out for 8 months, 656 Tenuipalpidae, 205 Tydeidae, 96 Tetranychidae samples collected during the entire study in an unsprayed coffee (n=3) do not generally make much sense due to very low mite populations. Similarly, the  6252 Eriophyidae, 207 Phytoseiidae, and 257 Tenuipalpidae gathered on IPS during the course of the experiment indicate that the species do not represent an meaningful mite population in the agro-ecosystem in which the experiment was conducted. However, the determination that IPS plants have the potential to serve as a harbor for predatory mites makes the study meaningful up to a point.

- In the abstract of the manuscript (MS), no results were given on the basis of species regarding the plants used as intercropped, and it would be appropriate to present specific data on them.

- In the introduction part of the MS, it is necessary to briefly explain why these plant species were selected by giving basic information about the species used as intercropped plants.

- Regarding the Material and Method, why weren't all three intercropping plant species given an equal number of places in each row? Clarification on this issue is required.

- Due to the fact that the experiments were set up in different areas (ca 200 m), evaluating each of the replications separately and presenting the results in a comparative manner would make the study more meaningful. Thus, the hypothesis put forward by the authors would have a chance to be tested more accurately.

- Why dry and rainy seasons were not considered as a factor that could affect the experiment results. In addition, why only 5 samplings were carried out, even though the trials were conducted for a period of 8 months.

- Depending on the experiment design given in Figure 2, what was the reason for sampling coffee plants located for every 4 m? For example, why wasn't a coffee plant located in every 2 m selected for sampling? If it is desired to monitor mite movements in real terms, keeping the distance short would be more appropriate for obtaining accurate data.

- Taking and counting samples in various ways at various stages of sampling creates confusion. This topic complicates the full understanding of this part of the Material method. In this section, the authors seem to have decided on their own experience without any reference, and it is a matter of debate how these changes in sampling and counting will affect the results. (For instance Lines 135-141).

- The statistical analysis carried out regarding the findings obtained are deemed appropriate for the purpose.

- It is an expected result that the abundance of herbivores is higher in line with the mite feeding behaviors. It should not be overlooked that this result is mainly due to the relative excess in eriopyhid density. This issue can be included in the discussion.

- In addition, what is the opinion of the authors for the practical use of the plants recommended to be used as intercropped in coffee plantations that are already cultivated?

Author Response

- The data and interpretations, however, could have been more informative if the experiments had been repeated for at least one more year to improve understanding and consistency of the findings. Therefore, I think that the data collected is moderately sufficient to support the main hypothesis of the authors. Related to this, considering that the study was carried out for 8 months, 656 Tenuipalpidae, 205 Tydeidae, 96 Tetranychidae samples collected during the entire study in an unsprayed coffee (n=3) do not generally make much sense due to very low mite populations. Similarly, the  6252 Eriophyidae, 207 Phytoseiidae, and 257 Tenuipalpidae gathered on IPS during the course of the experiment indicate that the species do not represent an meaningful mite population in the agro-ecosystem in which the experiment was conducted. However, the determination that IPS plants have the potential to serve as a harbor for predatory mites makes the study meaningful up to a point.

- In the abstract of the manuscript (MS), no results were given on the basis of species regarding the plants used as intercropped, and it would be appropriate to present specific data on them.

Reply: Done (line 20 - 22).

We added information about the mites found in the system, but we were unable to present much specific data due to word abstract limitations of the journal's rules.

- In the introduction part of the MS, it is necessary to briefly explain why these plant species were selected by giving basic information about the species used as intercropped plants.

Reply: Done (line 53 - 59).

- Regarding the Material and Method, why weren't all three intercropping plant species given an equal number of places in each row? Clarification on this issue is required.

Reply: Done (line 112 - 113).

As we described in “Study area”, each plot had 1080 m² (36 x 30m), with each coffee row apart from each other each 4 meters. The IPS were inserted in two lines of 30 linear meters, at the border of the plot. Thus, the amount of IPS was determined according to the limitation of space in each plot and the size of the plants, resulting in different quantities. As the trees are bigger than the shrubs, Inga and Senna have a smaller number of plants per area compared to Varronia.

- Due to the fact that the experiments were set up in different areas (ca 200 m), evaluating each of the replications separately and presenting the results in a comparative manner would make the study more meaningful. Thus, the hypothesis put forward by the authors would have a chance to be tested more accurately.

Reply: The three plots were not evaluated selapately because they are replicates. The three plots are equal: they have the same plant species and distances, replicated in same time and space. So, the replicates show that the results were not obtained by chance.

- Why dry and rainy seasons were not considered as a factor that could affect the experiment results. In addition, why only 5 samplings were carried out, even though the trials were conducted for a period of 8 months.

Reply: Evaluating the influence of climatic conditions was not the objective of our work. We did 5 samplings for logistical reasons, taking into account our time limitation and that the study was carried out during the covid 19 pandemic period, which limited us.

- Depending on the experiment design given in Figure 2, what was the reason for sampling coffee plants located for every 4 m? For example, why wasn't a coffee plant located in every 2 m selected for sampling? If it is desired to monitor mite movements in real terms, keeping the distance short would be more appropriate for obtaining accurate data.

Reply: The area was planned according to conventional coffee practices, considering that in the crops, tractors need to pass through each row to fertilize, pulverize and/or to pick the grains. That is why the space needed between rows is at least 4 meters. Thus, to analyze the effect of the distance of inserted plants in the experiments, the minimum possible between each sampling is 4 m, since the space between each row is 4 m.

- Taking and counting samples in various ways at various stages of sampling creates confusion. This topic complicates the full understanding of this part of the Material method. In this section, the authors seem to have decided on their own experience without any reference, and it is a matter of debate how these changes in sampling and counting will affect the results. (For instance Lines 135-141).

Reply: Although different amounts of leaves were sampled, we maintained a consistent identification patter of the material, identifying the same percentage of mites in all samplings, not affecting the results.

- The statistical analysis carried out regarding the findings obtained are deemed appropriate for the purpose.

- It is an expected result that the abundance of herbivores is higher in line with the mite feeding behaviors. It should not be overlooked that this result is mainly due to the relative excess in eriopyhid density. This issue can be included in the discussion.

Reply: Done (line 347 - 348).

- In addition, what is the opinion of the authors for the practical use of the plants recommended to be used as intercropped in coffee plantations that are already cultivated?

Reply: Done (line 323 - 325).

Round 2

Reviewer 3 Report

In general, recommendations made in the previous version of the manuscript were realized by the authors. In this context, the “abstract”,  “Introduction” “Material and Methods” and some part of the “discussion” sections have been revised in accordance with the recommendations.

I would like to inform you that the manuscript can be accepted for publication in the journal, with the final version submitted based on the referees' recommendations. 

Best regards